



# Biomass Burning Aerosol Heating Rates from the ORACLES 2016 and 2017 Experiments

Sabrina P. Cochrane[1,2], K. Sebastian Schmidt[1,2], Hong Chen[1,2], Peter Pilewskie[1,2], Scott Kittelman[1], Jens Redemann[3], Samuel LeBlanc[4,5], Kristina Pistone[4,5], Michal Segal Rozenhaimer[4,5,6], Meloë Kacenelenbogen[5], Yohei Shinozuka[5,7], Connor Flynn[3], Rich Ferrare[8], Sharon Burton[8], Chris Hostetler[8], Marc Mallet[9], Paquita Zuidema[10]

[1]Department of Atmospheric and Oceanic Sciences, University of Colorado, Boulder, 80303, USA

[2]University of Colorado, Laboratory for Atmospheric and Space Physics, Boulder, 80303, USA

[3]School of Meteorology, University of Oklahoma, Norman, Oklahoma, 73019, USA

[4]Bay Area Environmental Research Institute, Mountain View, 94035, USA

[5]NASA Ames Research Center, Mountain View, 94035, USA

[6]Department of Geophysics and Planetary Sciences, Porter School of the Environment and Earth Sciences, Tel-Aviv University, Tel-Aviv, Israel

[7]Universities Space Research Association/NASA Ames Research Center, Mountain View, 94035, USA

[8]NASA Langley Research Center, Hampton, VA, 23666, USA

[9]Centre National de Recherches Météorologiques, UMR3589, Météo-France-CNRS, Toulouse, France

[10]Department of Atmospheric Sciences, Rosenstiel School of Marine and Atmospheric Science, University of Miami, Miami, FL, 33149, USA

*Correspondence to*: Sabrina P. Cochrane (sabrina.cochrane@colorado.edu)

**Abstract.** Aerosol heating due to shortwave absorption has implications for local atmospheric stability and regional dynamics. The derivation of heating rate profiles from space-based observations is challenging because it requires the vertical profile of relevant properties such as the aerosol extinction coefficient and single scattering albedo (SSA). In the southeast Atlantic, this challenge is amplified by the presence of stratocumulus clouds below the biomass burning plume advected from Africa, since the cloud properties affect the magnitude of the aerosol heating aloft, which may in turn lead to changes in the cloud properties and life cycle. The combination of spaceborne lidar data with passive imagers shows promise for future derivations of heating rate profiles and curtains, but new algorithms require careful testing with data from aircraft experiments where measurements of radiation, aerosol and cloud parameters are better collocated and readily available.

In this study, we derive heating rate profiles and curtains from aircraft measurements during the NASA ObseRvations of CLouds above Aerosols and their intEractionS (ORACLES) project in the southeastern Atlantic. Spectrally resolved irradiance measurements and the derived column absorption allow for the separation of total heating rates into aerosol and gas (primarily water vapor) absorption. The nine cases we analyzed capture some of the co-variability of heating rate profiles and





their primary drivers, leading to the development of a new concept: The Heating Rate Efficiency (HRE; the heating rate per

unit aerosol extinction). The HRE, which accounts for the overall aerosol loading as well as vertical distribution of the aerosol

layer, varies little with altitude as opposed to the standard heating rate. The large case-to-case variability for ORACLES is

significantly reduced after converting from heating rate to HRE, allowing us to quantify its dependence on SSA, cloud albedo,

and solar zenith angle.

## 1 Introduction

Off the west coast of Southern Africa, a semi-permanent, seasonal stratocumulus cloud deck occurs in a broad region of

subsidence in the southeast Atlantic Ocean (Zuidema et al., 2016; Gordon et al., 2018). A westward outflow from the African

continent transports biomass burning (BB) aerosols out over the low clouds. The BB aerosols originate from fires in the

continental interior, where they have been lofted through convective heating (Zuidema et al., 2016; Adebiyi and Zuidema,

2018).

Due to flaming burning conditions, the BB aerosols have a high black carbon content, contributing up to 10% of the total

aerosol mass (Redemann et al., 2021; Dobracki et al., in prep.). With a single scattering albedo (SSA) of approximately 0.83

± 0.03 at 550 nm (Cochrane et al., 2021), these aerosols absorb a significant portion of the incoming and reflected radiation.

The direct interaction with radiation by scattering and absorption is known as the direct aerosol radiative effect (DARE).

Aerosol absorption induces diabatic heating of the layer, which has important consequences for atmospheric stability and

dynamics. The free tropospheric layer containing the aerosol is warmed while the surface below is cooled, which stabilizes the

low cloud deck. Observations indicate lower cloud top heights, higher liquid water paths and optically thicker, brighter clouds

when more shortwave-absorbing aerosols are present (e.g., Hansen et al. 1997; Dubovik et al., 2000; Kaufman et al., 2002;

Johnson et al. 2004; Wilcox 2010, Wilcox 2012, Yamaguchi et al., 2015; Zuidema et al., 2016, Gordon et al., 2018). If the

aerosols directly interact with the cloud below, the aerosol warming may facilitate cloud dissipation by reducing the relative

humidity, along with altering the cloud microphysics (Hill et al., 2008; Diamond et al., 2018; Zhang and Zuidema, 2019;

Mallet et al., 2020).

The aerosol layer in the southeast Atlantic is also colocated with elevated levels of water vapor (Haywood et al., 2003; Adebeyi

et al., 2015; Deaconu et al., 2019; Pistone et al., 2021), which have radiative impacts distinct from those of the aerosol layer.

In addition, water vapor can also modulate the aerosol itself: Magi and Hobbs (2003) find that the aged smoke (older than 45

minutes) in the southeast Atlantic is swollen (humidified) due to the water vapor.

The first step in understanding subsequent dynamic effects and cloud interactions is to determine heating rate profiles

throughout the aerosol layer. In addition to the information required to determine DARE (i.e., aerosol and cloud properties),

calculating heating rate profiles also requires accurate, detailed information on their vertical distribution.



Lidar instruments are specifically designed to penetrate through the atmosphere to obtain vertical profiles of atmospheric
backscatter and extinction, including those specific to aerosols. When positioned on satellites, lidar instruments can provide
aerosol extinction profiles for large spatial regions, which can be combined with additional observations that provide aerosol
optical properties (e.g., Deaconu et al., 2019), or they can be incorporated into climate models (Mallet et al., 2019; Tummon
et al., 2010; Gordon et al., 2018; Adebiyi et al., 2015; Wilcox 2010).

With current backscatter lidars, extinction profiles cannot be obtained directly due to the convolution of aerosol backscattering
and extinction in the returned lidar signal. For example, the Cloud-Aerosol Lidar with Orthogonal Polarization (CALIOP)
aboard the Cloud-Aerosol Lidar and Infrared Pathfinder Satellite Observation (CALIPSO) satellite measures the attenuated
backscatter coefficients at 532 and 1064 nm, from which the extinction coefficient is typically retrieved based on modeled
values of the extinction-to-backscatter ratio (also called the lidar ratio) inferred for each detected aerosol layer (Young and
Vaughan, 2009). The depolarization ratio method (DRM) (Hu et al., 2007; Chand et al., 2009; Deaconu et al., 2017;
Kacenelenbogen et al., 2019) also known as opaque water cloud (OWC) method takes advantage of the CALIOP
depolarization measurements and the transmission constraints provided by underlying low liquid water clouds to derive
accurate measurements of the above cloud aerosol optical depth (ACAOD) and the mean lidar ratio of the aerosol layer above
the cloud (Liu et al., 2015).

High Spectral Resolution Lidars (HSRL) are likely to be included in future space architectures. Among other advantages, they
will provide extinction profiles directly. This would be useful with the new concept of Heating Rate Efficiency (HRE) first
introduced in this paper (Section 5). It represents the heating rate at any altitude as obtained from aircraft observations per
aerosol extinction coefficient, which, by contrast to the heating rate itself, varies little throughout the profile. The HSRL-
derived extinction profiles could be directly translated into aerosol heating rates for regions where HRE is available, bypassing,
and thus directly constraining, radiative transfer calculations.

To develop the HRE concept, we use a combination of in situ and remote sensing observations from two years of the
ORACLES aircraft experiments (2016 and 2017). The aircraft is ideally suited for obtaining vertical information, which can
be taken from varied instruments for different parameters. In situ measurements can provide profiles of water vapor while
aerosol extinction profiles can be provided by either overflights with the High Spectral Resolution Lidar 2 (HSRL-2) (Burton
et al., 2018) or vertical profiles with the Spectrometer for Sky-Scanning, Sun-Tracking Atmospheric Research (4STAR;
Dunagan et al., 2013; LeBlanc et al., 2020) instrument.

For 9 cases (corresponding to those presented in Cochrane et al., 2021), we attribute the total heating of the layer to
contributions from the aerosol, water vapor, and other atmospheric gases. We also examine the dependence of the aerosol
heating rates and HRE on aerosol properties and cloud albedo – the same parameters that also modulate DARE (Cochrane et
al., 2021). Section 2 of this paper describes the data and general methods of the heating rate calculations. Section 3 provides
the results and discussion for heating rates segregated by absorber, while section 4 describes heating rate results along an
aircraft flight leg, known as a heating rate curtain. Section 5 introduces the new HRE concept and section 6 provides a
summary.



## 2 Methods

### 2.1 Data

The NASA ORACLES project utilized the NASA P-3 aircraft for the 2016, 2017, and 2018 deployments, while the ER-2 aircraft was additionally deployed in 2016 to record high altitude measurements (Redemann et al., 2020). Both aircraft were equipped with instrumentation to sample clouds and aerosols. The P-3 payload included both remote sensing and in situ instruments, while the ER-2 carried only remote sensing instrumentation.

To determine heating rates, we use a combination of measurements taken from the Solar Spectral Flux Radiometer (SSFR),
4STAR and HSRL-2. SSFR is a system consisting of two pairs of spectrometers that measure the upwelling (nadir) and downwelling (zenith) irradiance between 350 and 2100 nm. The zenith light collector is mounted on an Active Leveling Platform (ALP), which keeps the light collector level with true horizon during flight. SSFR is radiometrically calibrated pre- and post- mission, with field calibrations performed throughout each deployment to keep track of instrument changes throughout the duration of the experiment (Cochrane et al., 2019). The 4STAR instrument measures aerosol optical depth
(AOD) above the aircraft between 350 nm and 1650 nm, as well as provides column gas retrievals such as water vapor and ozone, aerosol intensive properties and cloud properties (Segal-Rozenhaimer et al., 2014; Pistone et al., 2019; LeBlanc et al., 2021). The instrument is calibrated pre- and post- mission using the Langley method at the Mauna Loa Observatory (e.g., Schmid and Wehrli, 1995; LeBlanc et al., 2020). HSRL-2 provides vertical profiles of aerosol backscatter and depolarization at 355 nm, 532 nm, and 1064 nm wavelengths, along with aerosol extinction at 355 nm and 532 nm wavelengths, all measured
below the aircraft (Hair et al., 2008; Burton et al., 2018). In 2016, HSRL-2 was mounted on the ER-2 but transitioned onto the P-3 for 2017 and 2018.

A major benefit of aircraft campaigns is the flexibility to perform distinctive flight maneuvers in a way that optimizes measurements and retrievals for different instrumentation. In this work, we use two distinct flight patterns: stacked legs (so-called radiation walls) and vertical profiles (so-called square spirals) to derive a) heating rate curtains and b) heating rate
profiles segregated by absorber, respectively. Figure 1 shows the locations of the radiation walls and spirals used in this study. Radiation walls, the traditional maneuver for radiation science flights, consist of vertically stacked legs flown in sequence along a fixed ground track. The stacked legs bracket the aerosol layer above and below, at the bottom of the layer (BOL) and at the top of the layer (TOL). For ORACLES, the BOL leg was located just below the aerosol layer and just above the cloud layer. The column AOD (spectral), ozone, and water vapor are measured by 4STAR, and spectral scene albedo is measured
by SSFR. The TOL leg is above the aerosol layer and the cloud, from which HSRL-2 measures profiles of extinction at 532 nm. In addition to the BOL and TOL legs, several other legs were flown within the wall, for example below and within the cloud, and within the aerosol layer. A full radiation wall often required over an hour to complete, and a square spiral was often included at one end of the radiation wall.

Since the radiation walls provide broad spatial coverage and allow us to sample varying cloud albedos and aerosol extinction
profiles for the same aerosol plume, we use the radiation wall data to understand the impact of scene variability (i.e. aerosol



loading and cloud albedo variability) on the aerosol heating rates. We do that by calculating heating rate *curtains* along the linear flight path, which require spiral measurements to be used in conjunction with AOD and scene albedo measurements from radiation walls (Table 3).

The downside of the radiation wall sampling is that it does not provide measurements throughout the aerosol layer. The new

square spiral maneuver, described in detail in Cochrane et al. (2019), allows for multiple measurements to be taken throughout the vertical profile over a short time period (10 - 20 minutes depending on the vertical extent of the aerosol layer). From the measurements in the column (the layer that encompasses both the aerosol and the water vapor layer) absorption can be derived, and ultimately heating rates segregated by absorber (primarily water vapor and aerosols, detailed in section 2.3).

Having the full column of measurements available from the spiral profiles led to the development of an aerosol optical property

retrieval (SSA and the asymmetry parameter (g), as detailed in Cochrane et al., 2021) that is directly tied to the irradiance measurements throughout the column. It is inherently difficult to retrieve these properties since the aerosol radiative effects can be relatively small compared to the horizontal variability of cloud albedo. The spiral sampling strategy, however, reduces cloud and aerosol inhomogeneity effects while maintaining correlation of measured irradiances throughout the spiral to the ambient cloud field (Cochrane et al., 2019).

From all useable spiral maneuvers from ORACLES 2016 and 2017 (Cochrane et al., 2021), we combine the retrieved aerosol properties with the in situ measured water vapor content, 4STAR-derived spectral extinction (derived from the derivative of AOD along the vertical) and atmospheric gas retrievals (Table 2). From these inputs, we calculate the vertical heating rate profiles for these cases. Since they are constrained by the total column absorption measured by SSFR, the heating rates are directly tied to the observations, ensuring consistency between aerosol properties, the water vapor profile, and the combined

radiative effects of the atmospheric constituents (radiative closure). Since this is done spectrally, we can segregate by major absorbers to determine their relative contributions to the overall heating.

## 2.2 Heating rate calculations

To calculate heating rates for both the spiral profiles and radiation walls, we use the 1-dimensional (1D) radiative transfer model (RTM) DISORT 2.0 (Stamnes et al., 2000) with SBDART for atmospheric molecular absorption (Ricchiazzi et al.,

1998) within the libRadtran library *(*Emde et al., 2016; libradtran. org). Table 1 lists the input parameters and their sources required for the calculations. The general method is to segregate the absorption by strategically eliminating a single constituent for separate calculations (Kindel et al., 2011). First, we calculate the total heating rate from cloud top to well above the top of the aerosol layer, then remove a single atmospheric component from the profile (e.g., aerosols) and calculate the heating rate again. The difference between the two calculations provides the isolated heating rate for the individual component that was

excluded in the second calculation.

The radiative transfer calculations are performed in this manner rather than calculating the heating rate of each component directly in order to maintain physical consistency. As incoming radiation travels through the atmosphere, less and less total radiation reaches lower altitudes. When there is a strong absorber present, such as an aerosol layer, the decrease of radiation



with altitude is amplified since the layer absorbs a significant portion of the incoming radiation. Calculating absorption (and
heating) for a single component directly does not consider the attenuation by other constituents, potentially leading to an
overestimation of the heating rates at altitudes below that of the absorbing layer.

We calculate the heating rate of a layer following the equation presented in Schmidt et al. (2021):

$$\frac{\Delta T}{\Delta t} = \frac{1}{\rho c_p}\frac{\Delta F_{net}}{\Delta z} = \int \frac{1}{\rho c_p}\frac{\Delta F_{net,\lambda}}{\Delta z}\, d\lambda, \tag{1}$$

where $\rho$ is the density, $c_p$ is the constant-pressure specific heat capacity of air, $\Delta z$ is the layer thickness, and $\Delta F_{net}$ is the
difference of the net irradiance at the layer top and bottom, i.e., the absorbed irradiance in that layer. Since the absorbed
irradiance is a spectral quantity, the different wavelengths contribute varying amounts (Fig. 3) and must be integrated to find
the total heating rate at each altitude. The heating rate is calculated at each layer altitude defined in the model (every 0.2 km
for cloud top altitude (approximately 1 km) < altitude < 7km; every 1 km for 7 km < altitude < 12 km).

The RTM requires the spectral aerosol optical properties SSA and the asymmetry parameter (g) along with the vertical profile
of aerosol extinction and the spectral albedo. SSA and g spectra are taken from the SSFR retrievals presented in Cochrane et
al. (2021) and assumed to be vertically homogenous since the retrieved values represent the entire layer. The spectral albedo
measured by SSFR just above the cloud top defines the "surface" beneath the aerosol layer at the altitude of the cloud top.
Aerosol extinction is taken either from HSRL-2 (for radiation walls) or derived from the 4STAR AOD profile (for spirals) and
detailed in section 2.3 and 2.4.

Atmospheric gases are defined by the standard tropical atmosphere included in the libRadtran package (Anderson et al., 1986).
The water vapor and ozone profiles, however, are modified to reflect the specific conditions encountered during the time of
the measurements. Specifically, due to the consistently high water vapor observed in the biomass burning plume (e.g., Pistone
et al 2021), the vertical distribution of the water vapor within the aerosol layer is first determined by in situ measurements
taken by the P-3 hygrometer during the spiral profiles. Beyond the top of the aerosol layer (TOL), the values revert to those
found in the standard atmosphere. The full column of water vapor (the water vapor path) is then scaled such that the total
column (between BOL and top of atmosphere (TOA)) is equal to the 4STAR retrieval at the lowest flight altitude. One example
profile is shown in Figure 2. Similarly, the ozone profile is scaled to the 4STAR ozone retrieval measured at the BOL. For the
spiral cases, the water vapor and ozone profiles are consistent with those used within the aerosol retrievals of Cochrane et al.,
2021.

**3 Heating Rate Segregation by Absorber**

The vertical profiles measured during the spiral flight patterns allow us to separate heating rates for different constituents
beyond that of just the aerosol. This separation process relies on the principle that different atmospheric gases, aerosols, and
water vapor absorb radiation in different wavelength ranges.



To separate the total heating into component-specific heating rates, we strategically remove components one by one and re-run the radiative transfer calculations. For example, to determine the aerosol heating rate profile, we set the aerosol extinction profile to zero and calculate the heating rate profile. The difference between the total heating rate (where all constituents are included: the aerosol layer, water vapor, and ozone (all from measurements) as well as the libRadtran standard carbon dioxide and oxygen) and the total heating rate spectrum with the aerosol layer removed is the aerosol heating rate profile. The same technique is applied to each other constituent. Figure 3 illustrates the different heating rate spectra between the different components at one example altitude.

The aerosol extinction profile at each wavelength is obtained from the 4STAR measurements (derivative of AOD with respect to altitude), which provide the required spectral dependence throughout the profile. This is only possible when the profile of AOD measurements is available (i.e., from a spiral). This approach is consistent with the methodology used to retrieve the aerosol optical properties (SSA and g); Cochrane et al., 2019; Cochrane et al., 2021). Data conditioning is required such that the AOD profile decreases monotonically with altitude to eliminate any unphysical (negative) extinction values. These extinction profiles are consistent with those used in the aerosol property retrieval and have a 20 m vertical resolution; one example profile is shown in Figure 2. The directly retrieved HSRL-2 aerosol extinction profile at 532 nm is close to the 4STAR derived aerosol extinction profile. Any differences are due to the different location where these were acquired (wall vs. spiral), or due to the data conditioning applied to the 4STAR data.

Table 2 presents the input values for the RTM heating rate calculations required input parameters for each of the spiral cases..

**3.1 Heating rates from spiral profiles**

Total, aerosol, and water vapor heating rate profiles separated by year are presented in Figures 4a, b, and c. For the selected cases analyzed here, the aerosol plume was lower in altitude in 2017 than 2016, which can be seen in Figures 4a (total heating rates) and 4b (aerosol heating rates). The peak aerosol heating rates were similar in both years, approximately 4-6 K day$^{-1}$, with the exception of one 2017 case for which the aerosol loading was significantly higher than other cases.

In 2016, the aerosol and water vapor layer were generally collocated in altitude (Figures 4b and 4c), supporting the common assumption that aerosol and water vapor advect jointly from the continent. In 2017, by contrast, the primary aerosol loading resides below the free tropospheric water vapor loading. The maximum water vapor heating (excluding the boundary layer) was between 2-4 K day$^{-1}$ at the specified solar zenith angles in both 2016 and 2017, though the 2017 water vapor heating rate peak is much broader than in 2016. These differences are possibly due to the differing locations and meteorological conditions, examined further in a campaign-wide analysis by Pistone et al., 2021.

Compared to Deaconu et al. (2019), who derived instantaneous heating rates from MODIS/POLDER/CALIOP/ECMWF satellite instrumentation from June-August 2008, we generally find lower peak aerosol heating rates: 4-6 K day$^{-1}$ versus 9 K day$^{1}$. This is likely due to differences in aerosol loading and vertical structure, since Deaconu et al. 2019 analyzed data much closer to the continent than any of our ORACLES cases. The instantaneous water vapor heating rate estimates, albeit at different solar zenith angles, are consistent: we calculate 2-4 K day$^{-1}$ compared to 3 K day$^{-1}$.



The water vapor contributions reported in this work as well as in Deaconu et al. (2019) are significantly larger than those in Adebiyi et al. (2015), who find that the maximum of the instantaneous water vapor SW heating rate is only ~10% of the maximum aerosol heating rate for clear sky near the island of St. Helena averaged over September-October: approximately

0.12 K day$^{-1}$ (water vapor) relative to 1.2 K day$^{-1}$ (aerosol). By contrast, our averaged maxima show the water vapor heating rate to be ~60% of the aerosol heating rate: 2.8 K day$^{-1}$ compared to 4.6 K day$^{-1}$. One reason is that the water vapor heating rates reported in Adebiyi et al. (2015) represent the difference between composite humidity profiles constructed from days with light and heavy aerosol loadings (AOD<0.1 and AOD>0.2 respectively), and rely on the positive correlation between aerosol and water vapor.

Comparing our results to other observations or model estimates previously reported in the literature (e.g., Marquardt Callow et al., 2020; Baró Pérez et al., 2021) is also challenging because heating rates depend on numerous parameters beyond spectral extinction profiles and single scattering albedo: most importantly on the albedo of the underlying scene, layer depth and vertical resolution, as well as on variations in location (aerosol type) and solar zenith angle. Also, some studies report column-averaged, other than peak heating rates. This challenge is one of the motivations for the development of the heating rate *efficiency*

(Section 5) which turns the heating rate (an extensive parameter) into an intensive quantity. If generally adopted, this new concept will allow for improved comparisons between studies.

Nevertheless, we first calculate the traditionally reported column-averaged heating rate (from cloud top to the top of the aerosol layer). Figure 5 shows the breakdown of the column-averaged total heating rate into its individual components, averaged for all cases. As expected, the largest contribution of heating stems from the aerosol (55.7% of the total), followed by that from

water vapor (37. 5%).

## 4 Heating Rate Dependence on Scene Parameters: Heating Rate Curtains

For the radiation walls, we calculate heating rate curtains to examine the relative dependencies of the aerosol heating rates on the AOD and the underlying albedo. This requires a) aerosol optical properties and water vapor profile (both obtained from a spiral retrieval and held constant), b) the SSFR-measured albedo spectra, the AOD spectra and column ozone retrievals from

4STAR, all measured from the BOL leg of the radiation wall, and c) aerosol extinction profiles at 532 nm measured by HSRL-2 from the TOL leg (Table 2). As mentioned above, radiation walls were typically accompanied by a spiral at one end. The spiral provides the aerosol optical properties along with the water vapor profile, while the collocated BOL and TOL legs provided the horizontal aerosol and cloud variability. Unfortunately, the spatial collocation is of little help since the time lag between sequential sampling of the TOL and BOL legs is large enough that the cloud field is likely to change. In addition,

some radiation walls included TOL and BOL layers that were not perfectly collocated and/or there was not an associated spiral that produced a valid aerosol retrieval.

Therefore, it was important to find a way to examine the heating rate dependence on the scene variability without knowing the albedo and AOD spectra underlying a *specific* aerosol extinction profile. We therefore assessed the dependence statistically





by considering 1) the albedo variability and 2) the AOD. For 1), we used the collection of measured spectra along the BOL

leg, regardless of whether the BOL and TOL legs were collocated. From that, we created five representative albedo spectra for the heating rate calculations: the minimum, maximum, mean, and mean ± one standard deviation of the data collection. For part 2, we pursued a similar approach with the HSRL-2 extinction profiles from the TOL leg, which provide the vertical distribution of the aerosol. We paired this with the 4STAR-derived AOD from the bottom of the spiral (or from the leg-averaged BOL retrievals if a spiral was not available) as the basis for spectral extrapolation, which maintains spectral

consistency across all heating rate calculations, as well as with the aerosol property retrievals. It should also be noted that for the two cases with no useable spiral, the aerosol optical properties were set to the ORACLES mean SSA and g spectra for the 2016 and 2017 deployments (Table 3 Cochrane et al., 2021) and the water vapor profile (and 4STAR column water vapor for scaling) are obtained from the closest spiral of the day regardless of whether there was a valid aerosol retrieval.

For each radiation wall, we calculate aerosol heating rates for extinction profiles from HSRL-2 along the TOL, approximately

one profile every 5 seconds (interpolated from the 10 second resolution reported in the HSRL-2 data product) which is approximately every 0.005 degree of latitude or longitude. We extend the 532 nm extinction measured by HSRL-2 to the entire spectrum using the representative 4STAR AOD spectra in the following manner:

1. Integrate HSRL-2 extinction $\beta_{ext}$ to get AOD at 532 nm:

$$AOD_{532}^{HSRL}(x) = \int_{BOL}^{TOL} \beta_{ext}(z,x)dz. \tag{2}$$

2. Compare $AOD_{532}^{HSRL}(x)$ (from TOL) with $AOD'^{4STAR}_{532}$ (where $AOD'^{4STAR}_{532}$ is at either the spectrum measured at the bottom of the spiral or the average spectrum of the BOL leg) and

rescale the entire 4STAR AOD spectrum by the ratio between them:

$$AOD'_\lambda = AOD'^{4STAR}_\lambda * \frac{AOD'^{4STAR}_{532}}{AOD_{532}^{HSRL}(x)}. \tag{3}$$

3. Take the rescaled AOD spectrum, divide by HSRL-2 AOD to get rescale factors at each wavelength ($R_\lambda = 1$ at 532 nm):

$$R_\lambda(x) = \frac{AOD'_\lambda}{AOD_{HSRL}(x)}. \tag{4}$$

4. Get extinction profiles at all wavelengths using the rescale factors:

$$\beta_{ext,\lambda}(x) = \beta_{ext,532}(x) * R_\lambda(x). \tag{5}$$

In this approach, we are making two implicit assumptions: 1) that the 532 nm extinction can be used as a proxy to the full extinction spectrum and 2) that the spectral shape of extinction does not vary with location or altitude. This may not be fully

representative of the aerosol layer: For example, there could be different amounts of coarse mode aerosol in various layers of the plume, and the SSA is not necessarily constant with altitude (LeBlanc et al., 2020; Redemann et al., 2021; Dobracki et al., 2021). One possible method to refine assumption 2 would be to introduce an altitude-specific spectral extinction adjustment based on the HSRL-2 Angstrom exponent product, which could be investigated in further studies.

Figure 6 shows one example of the aerosol heating rate curtains calculated using the HSRL-2 extinction profiles. The high

resolution of HSRL-2 allows us to resolve the minor variations within the aerosol plume while providing sufficient data to





analyze the aerosol layer statistically. For each case, the albedo, AOD, and aerosol extinction profile vary across the wall, while the intensive aerosol optical properties are assumed to remain constant. For this particular case, the thickest part of the aerosol layer occurs between 2km and 3km, with higher heating rates to the south. Although the AOD along the wall ranges only from 0.19 to 0.22 at 532 nm, the high variability in the albedo causes large differences in the heating rate values.

The heating rate curtains provide enough data to examine the heating rate dependence on both the AOD and albedo along the wall for each of the cases. Each plot in Fig. 7 shows a thick solid profile (black) for the total heating rate and one for the aerosol heating rate (gray). These profiles represent the mean along the entire wall of HSRL-2 extinction profiles at the mean SSFR albedo value. The overlaid error bars represent the standard deviation (1-sigma) of the heating rate due to variable extinction for the mean albedo value. The dashed lines represent the mean for all extinction profiles for the lowest albedo and the highest

albedo encountered during the wall.

In some cases, such as 20160920 (7a) and 20160924 (7b), the variation in AOD results in larger heating variation than caused by the changing albedo below the layer. In other cases, 20170813 (7c) and 20170824 (7d) the variation in heating due to changing albedo along the wall is greater than that introduced by the AOD. In the 20170826 (7e) case, the variation in albedo affects the total heating rate more than the AOD variation, while for the aerosol heating rate, the AOD and albedo variation

introduce approximately equal variations.

For every case, the mean total (aerosol) heating rate ranges between 4-8 K day$^{-1}$ (2-3 K day$^{-1}$). The aerosol heating values are similar to those found for the region by Keil and Haywood (2003; 2.3 K day$^{-1}$), Gordon (2018; 1.9 K day$^{-1}$), and Wilcox (2010; 2.0 K day$^{-1}$). In contrast, we find slightly larger aerosol heating rate values than Tummon (2010; 1 K day$^{-1}$) and Adebeyi et al. (2015; 1.2 K day$^{-1}$). Of course, we do not necessarily expect agreement with prior studies since there are numerous

differences between them such as different observational periods, locations, approach, and aerosol optical properties.

A direct comparison (in both observation location and time) for the September 24[th], 2016 radiation wall (Fig 7, panel f) between the ALADIN-Climate (Aire Limitée Adaptation dynamique Développement InterNational) regional model (Nabat et al., 2015) and our direct calculations show good agreement for the average aerosol shortwave heating rate profile. The peak heating values within the aerosol plume are consistent, though those from the ALADIN model are slightly higher in altitude. However,

the aerosol extinction profile from the ALADIN model is larger in magnitude than that of ORACLES. Likely, the difference in extinction is offset by other driving parameters of the heating rate (e.g., cloud albedo; aerosol optical properties), resulting in the good agreement between heating rates. The negative values between 7-10 km in the ALADIN profile may possibly come from differences in high-cloud properties between the two model simulations (Mallet et al., 2019).

**5 Heating Rate Efficiency**

Heating rate variability between cases, locations, and campaigns strongly depends on drivers such as extinction, cloud albedo, aerosol optical properties and the vertical distribution. Unlike the related quantity DARE, which is usually studied at a single level such as the top of atmosphere or the surface, heating rates extend throughout the entire column. In the past, it has been





difficult to condense this into a single value to report and compare, leading to varying definitions and reported values in the literature (e.g., peak heating rate, column average heating rate.)

The main drivers of the heating rates are key to understanding the origin of the variability and overcoming the factors that limit our ability to understand the heating rate variability. For aerosols, the AOD is the most significant driver for the overall heating rate value. In Fig 8a, the column-averaged aerosol heating rate for each spiral case is shown as a function of the AOD at 550 nm, with some cases labeled by their 550 nm SSA. As expected, aerosol loading, i.e., AOD, is the main driver of the heating rate, but a simple relationship is not expected because the vertical structure of the plume and its thickness very from case to

case. Still, the dependence can be confirmed with radiative transfer calculations. We calculate aerosol heating rates as a function of AOD (black dashed line in Fig. 8a) for SSA=0.83, g=0.54, and an albedo of 0.60 (values listed for 550 nm, the full mean ORACLES spectra are used within the RTM). The vertical profile for these calculations was set to that of the 20160920 #2 spiral case.

For water vapor, the main controller of the heating rate is the water vapor path (the layer-integrated water vapor content),

shown in Fig 8b. Each case is labeled by the aerosol SSA value at 550 nm to link back to Fig 8a. The deviation of the individual cases from a linear relationship cannot be explained by the scene albedo at the water vapor absorption bands since the dependence (not shown) is too weak. The most plausible explanation is therefore the vertical distribution of the water vapor. The only explanation is therefore the vertical distribution of the water vapor.

From Figures 8a and 8b, we can see that layer heating by water vapor dominates over aerosol-induced heating up to a mid-

visible aerosol optical thickness of about 0.25 (for a case with a water vapor path of ~1.2 g cm$^{-2}$, SSA=0. 83).

From Fig 8a, we can see that the variability in the AOD between cases overwhelmed any signal from the smaller heating rate drivers such as SSA and albedo. The problem is that the variability of AOD and its vertical distribution are intertwined. To isolate the vertical distribution of the aerosol extinction coefficient from its column integral, we divided the heating rate at any given altitude ($z$) by the aerosol extinction coefficient, rather than working with the column-averaged heating rate as in Fig. 8

and in previous studies. We named this "intensive" parameter the relative heating rate *efficiency*, borrowing from relative aerosol forcing efficiency (Redemann et al., 2006), which is also independent of the aerosol loading. The HRE is defined as:

$$HRE(z) = \frac{HR_{aerosol}(z)}{\beta_{ext}(z)} * \left(\frac{1}{F^{\downarrow}(z)}\right), \qquad \qquad e(6)$$

where $HR_{aerosol}$ is the aerosol heating rate profile, $\beta_{ext}$ is the aerosol extinction profile at 532 nm, and $F^{\downarrow}$ is broadband downwelling, spectrally integrated from 350nm - 2100 nm. The units of HRE are [K/day] / [1/km] / [W/m$^2$].

The normalization by the incident broadband irradiance at any given altitude was included in the definition to account for the changing SZA and self-dimming (related to the AOD) of the layer as we step further down into it. Using HRE instead of HR provides and alternate way to examine the dependence of aerosol heating on SSA, Albedo, and SZA. Figure 9 shows an example of the vertical profile of the aerosol heating rate, extinction, and HRE.

To examine the dependencies of the heating rate efficiency on its drivers, we used the extinction profile, SSA retrieval, and

measured cloud albedo for one spiral case (20160920 #2) as a reference and determined the HRE variation for changes in the



SSA, cloud albedo, and SZA. The SSA spectrum changes relative to the reference case are determined through the co-albedo, where the reference case is scaled to range from 0.01 to 0.24 at 532 nm. The scaled co-albedo (1-SSA) spectra are then translated back into SSA for input into the RTM. The cloud albedo spectrum changes relative to the reference case are obtained via cloud retrievals based on the reference albedo spectrum, and then varying the retrieved cloud optical thickness to modulate

the spectral albedo (for more details, please see Appendix A.3.2 in Cochrane et al., 2021.) While the SSA and cloud albedo were varied in the same way for the nine cases, the extinction profiles, water vapor profile remained case-specific.

Fig 10 shows the dependence of HRE on its main drivers. We can see that the HRE (in contrast to layer-averaged heating rate) varies little from case to case, and that its variability is tightly constrained by the albedo and SSA. It increases with increasing cloud albedo, consistent with the finding of Adebiyi et al. (2015). The same finding would be true for any bright surface (not

just a cloud), although more complicated spectral dependencies would have to be considered. The black dashed line indicates the mean fit line, which extends from albedo values 0 to 1. From the fit line, it is apparent that the HRE increases by approximately a factor of 2 across the full albedo range from 0 to 1. This can be understood intuitively, considering that the layer at an albedo of 1 is essentially illuminated "twice" – from the top and from the bottom, doubling its heating.

Fig 10b shows that the HRE decreases with increasing SSA. This is expected, since an increasing SSA indicates less absorption

(relative to scattering), leading to lower heating rates. The gray symbols (calculated for the widened SSA range for one case only) show that HRE goes to 0 as SSA goes to 1 (purely scattering, no absorption). Decreasing the SSA from 0.9 to 0.8 almost doubles the HRE, which is also expected. It should be emphasized that the clear and easy-to-interpret dependence of HRE on the albedo and SSA could not have been achieved with the previous concepts of layer-mean or maximum heating rate. However, the benefits of the HRE concept also have a limit in that the extinction, SSA and albedo all have a spectral

dependence, which is important to consider when deriving an inherently broadband quantity such as layer heating. In addition, one needs to consider the dependence of the heating rate on the solar geometry. Fig 10c shows that HRE increases as a function of SZA, with a sharp increase at the higher SZAs (lower sun elevations). This is due to the increase in path length through the aerosol layer as the sun moves towards to horizon.

Fig 10 demonstrates the utility of the HRE and highlights the effectiveness of the parameter for reducing the complexity of

heating rates. At the albedo value of 0.5, SSA of 0.85 and a 20 degree SZA (location of the black lines in Figure 10), the HRE is 0.025 K / day / km$^{-1}$ / W m$^{-2}$ with a standard deviation of only 0.002, approximately 8% in relative terms. This small variability shows HRE could be used to translate extinction profiles in the region directly into aerosol heating rates if mid-visible cloud albedo and SSA are also known. In other words, the variability in extensive parameters (e.g., extinction) is higher than intensive parameters (e.g., SSA, g) and therefore, regionally and seasonally defined HRE are useful. If available for a

specific region, the HRE concept would allow a direct translation from mid-visible extinction to heating rate. Of course, if SSA varies appreciably within the layer, that dependence may have to be made explicit. a significant reduction in complexity.



## 6 Summary and Discussion

Observations from the 2016 and 2017 ORACLES experiments allow us to introduce a method of determining heating rate profiles as directly as possible by linking the heating rates to SSFR-measured irradiance profiles. The spectrally resolved

irradiance measurements and the derived column absorption allowed the separation of heating rates by absorber, most importantly the separation of water vapor heating from aerosol heating. We found that for many cases, the water vapor heating rate is nearly as large as the aerosol heating rate, on average 38% compared to 56% of the total heating (respectively), highlighting the large influence of the atmospheric water vapor distribution on the total heating rate distribution – even for optically thick aerosol layers. We also found that layer heating by water vapor coincident with aerosol dominates over aerosol-

induced heating up to a mid-visible aerosol optical thickness of about 0.25 (for one case with a water vapor path of ~1.2 g cm$^{-2}$, SSA=0.83).

Analysing the dependence of the heating rate on the driving parameters (AOD, albedo, SSA) showed that the primary parameter affecting the aerosol heating rate is the AOD, and to a lesser extent the cloud albedo and aerosol SSA. For the mean SSA spectrum encountered during ORACLES (i.e., SSA=0.83 at 550 nm), the heating rate increases by ~0.5 K day$^{-1}$ per 0.1

increase in aerosol optical depth (for albedo=0.6 at 550 nm). The heating rate variability, however, is highly case dependent because of the co-variability of the driving parameters.

The vertical distribution of the aerosol layer in relation to the underlying cloud also introduces variability from case to case that cannot easily be evaluated. The impact of the vertical distribution makes heating rates more difficult to generalize than other radiative effects, such as the direct aerosol radiative effect (DARE, Cochrane et al., 2020). The new HRE parameter,

however, does show potential for such a generalization because it defines the heating rate per extinction coefficient. The HRE, in contrast to the heating rate, has a clear dependence on albedo and SSA that could not have been determined through either maximum or layer-averaged heating rate concepts. By reducing the complexity of the convoluted relationship between heating rates and its drivers, the HRE parameter makes it possible to investigate the relationship between heating rates and other parameters besides the aerosol loading and extinction profile. In the future, the HRE relationships established in this work

could be formalized into a general parameterization applicable for the ORACLES study region. However, it will be important to consider that this type of broadband generalization will require spectral dependencies of extinction, SSA, and albedo.

For one preliminary comparison between our calculations and the ALADIN regional climate model output, we found consistent peak aerosol heating rates. The heating rate curtains, calculated along radiation walls using HSRL-2 extinction profiles, can currently only be derived from aircraft observations. To arrive at a similar product from satellite retrievals, one would have

to ensure that the heating rates are consistent with a radiative flux constraint (at the very least at TOA), in addition to filtering out cloud inhomogeneity effects. This will be relevant for planned space-borne missions.





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

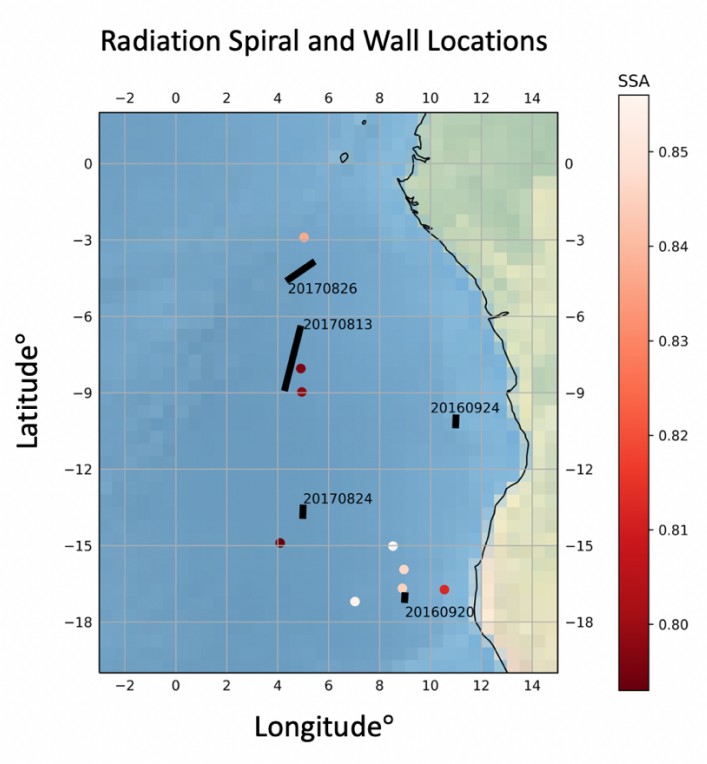

**Figure 1: The location of spirals and radiation walls within the ORACLES study region. Spirals are shown as circles colored by the SSA retrieval (Cochrane et al., 2021) at 550nm. Radiation walls are shown as black rectangles labeled by date.**





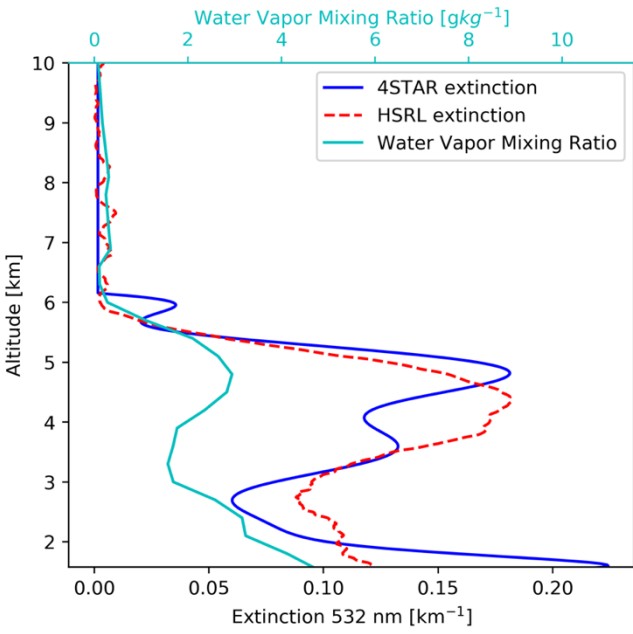


**Figure 2. 20160920 4STAR-derived 532 nm aerosol extinction profile (dark blue), averaged HSRL-2 532 nm aerosol extinction profile across the radiation wall (red dashed), and the water vapor content profile measured during the spiral profile (cyan). The 4STAR-derived extinction (532 nm) and the water vapor mixing ratio from the spiral profile is at a different location relative to the wall-averaged HSRL-2 extinction profile.**

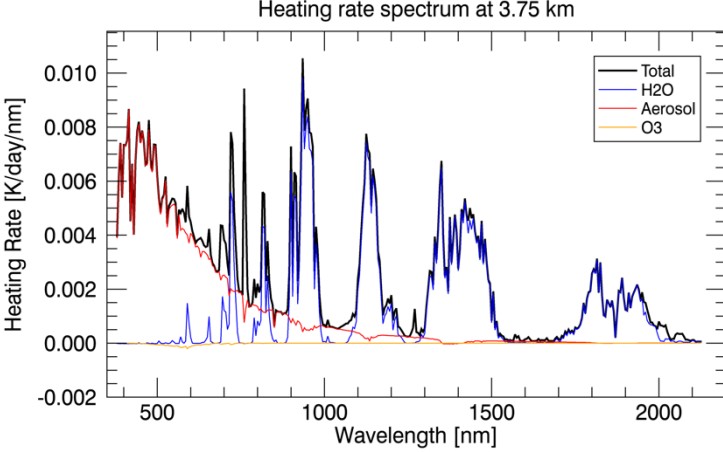


**Figure 3. The total heating rate spectrum (black) shown along with individual heating rate spectra for aerosol, water vapor, and ozone at 3.75 km.**





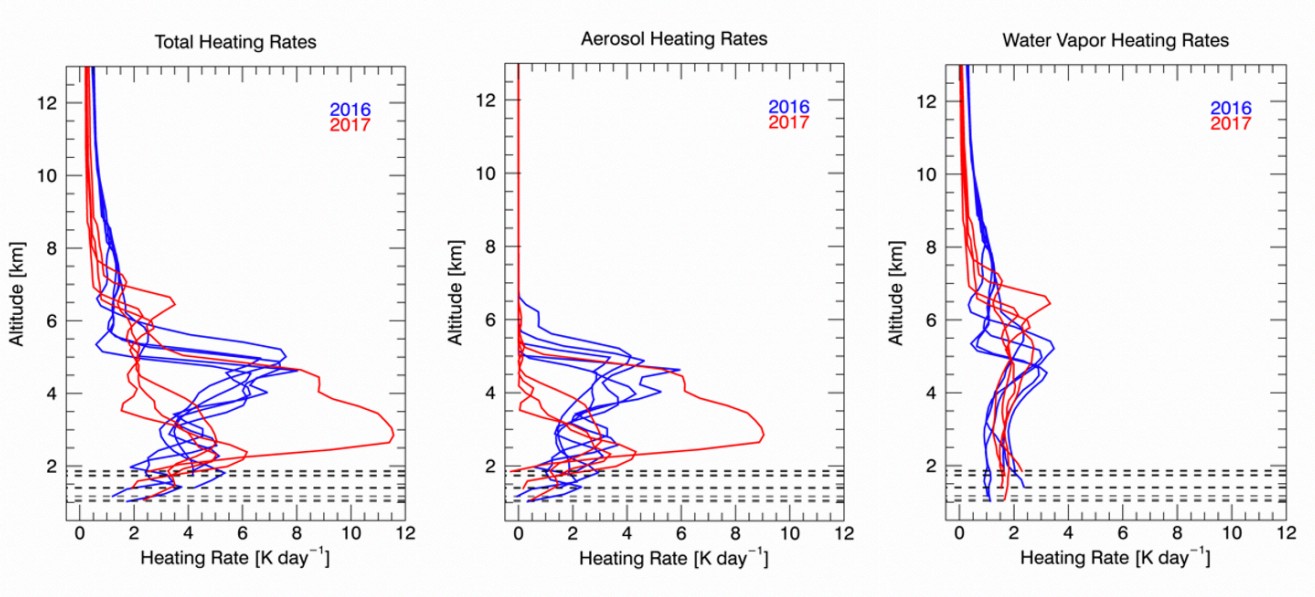

**Figure 4. 2016 (blue) and 2017 (red) vertical heating rate profiles for (a) total (b) aerosol and (c) water vapor calculated from the**
**spiral cases with valid aerosol SSA and g retrievals. Dashed lines indicate the bottom altitude for each profile.**

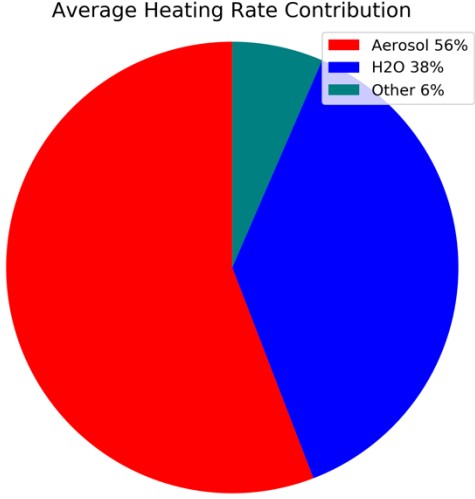

**Figure 5.  The case-average contributions to the total heating rate (column-averaged) from the nine spiral profiles mentioned in**
**Table 1. The heating rate contribution of the category labeled "Other" is primarily gas (ozone, oxygen, and carbon dioxide)**
**absorption.**



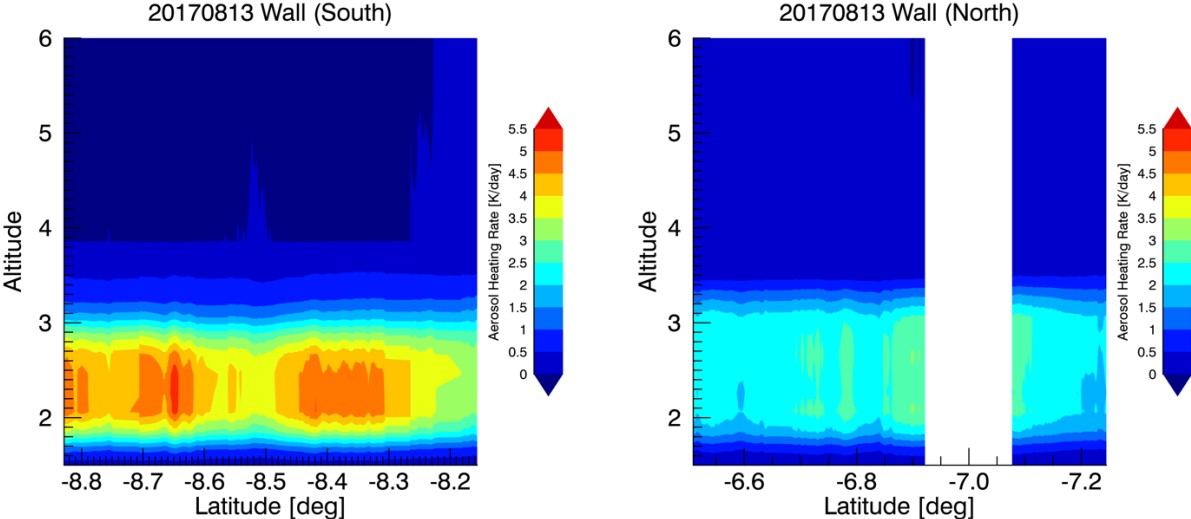

**Figure 6. Heating rate curtains calculated using HSRL-2 measured extinction for the 20170813 radiation wall (shown here in separate plots: North (right) and South (left)). Peak heating of ~5 K/day occurs between 2 and 3 km. The underlying albedo is significantly higher on the left plot than the right (i.e. further south); 0.55 compared to 0.19 at 532 nm contributing to higher aerosol heating rates. Missing results between -7.25 and -8.15 are due to in-cloud sampling that replaced above-cloud albedo measurements and serve as the break point between North and South ends of the wall.**




**Figure 7. Mean vertical profiles for total (black) and aerosol (gray) heating rates across the radiation walls for 5 cases (a, b, c, d, e,**
**605** **calculated from HSRL-2 extinction profiles and SSFR/4STAR SSA and g values. The red error bars represent the variability due to**
**changing extinction, while the dashed lines represent the variability due to changing albedo across the wall. The horizontal solid line**
**indicates cloud top height. f) Average aerosol heating rate profiles across the 20160924 radiation wall calculated from ORACLES**
**versus the ALADIN climate model.**





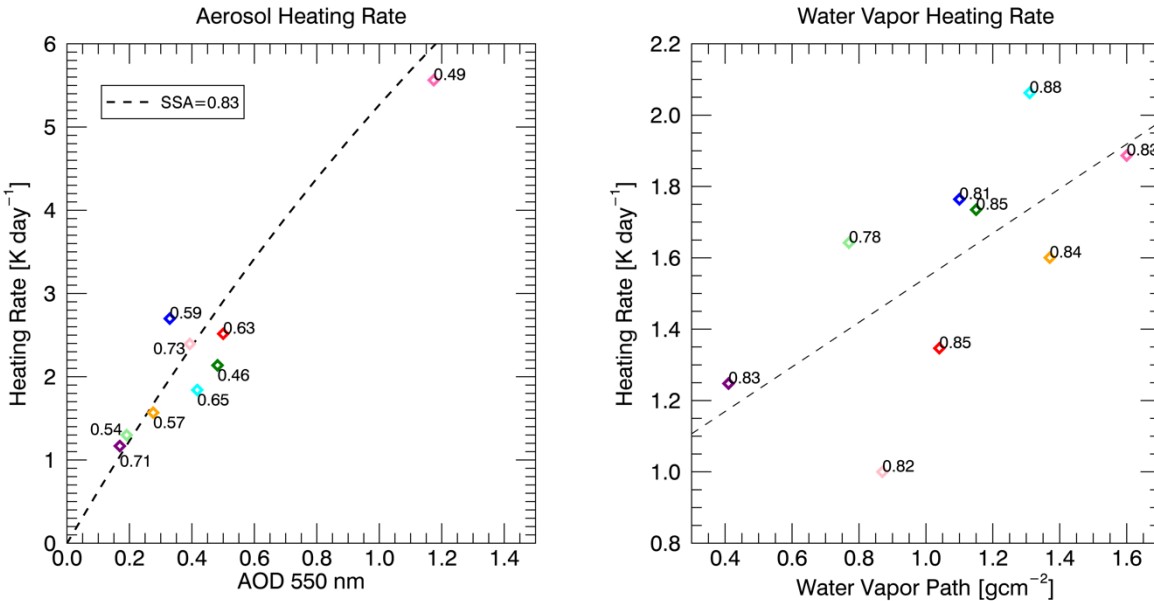

**Figure 8. a) Aerosol heating rate as a function of AOD at 550 nm. Column-averaged values from each spiral case are shown as colored points labeled by their 550 nm albedo value. The black dashed line indicates RTM calculations using mean SSA (0.83, 550 nm) and albedo (0.6, 550 nm) from all cases, and a range of AOD spectra (ranging from 0 to 1.4 at 550 nm). b) Water vapor heating rate as a function of the water vapor path. The grey dashed line is a simple linear fit to highlight the dependence. Cases are labeled by the 550 nm SSA value.**





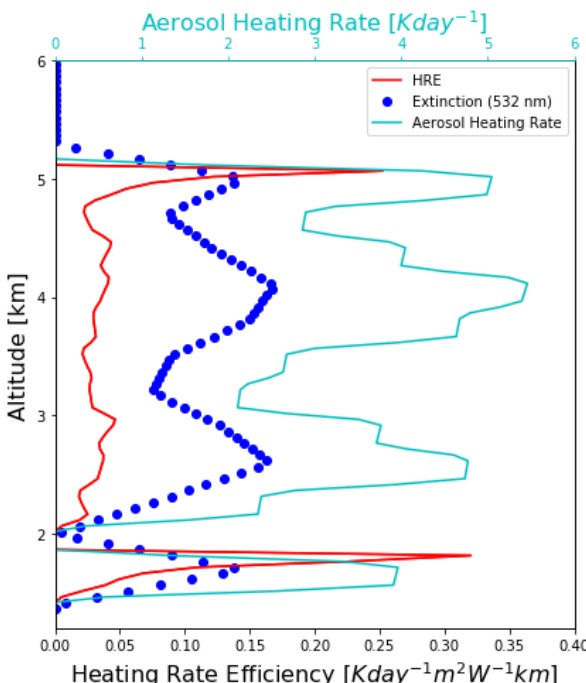

**Figure 9. Vertical profiles of HRE, aerosol heating rate, and aerosol extinction at 532 nm. The vertical resolution defined in the model for which these calculations are performed is set to every 0.05 km for altitude < 7km; every 1 km for 7 km < altitude < 12 km.**

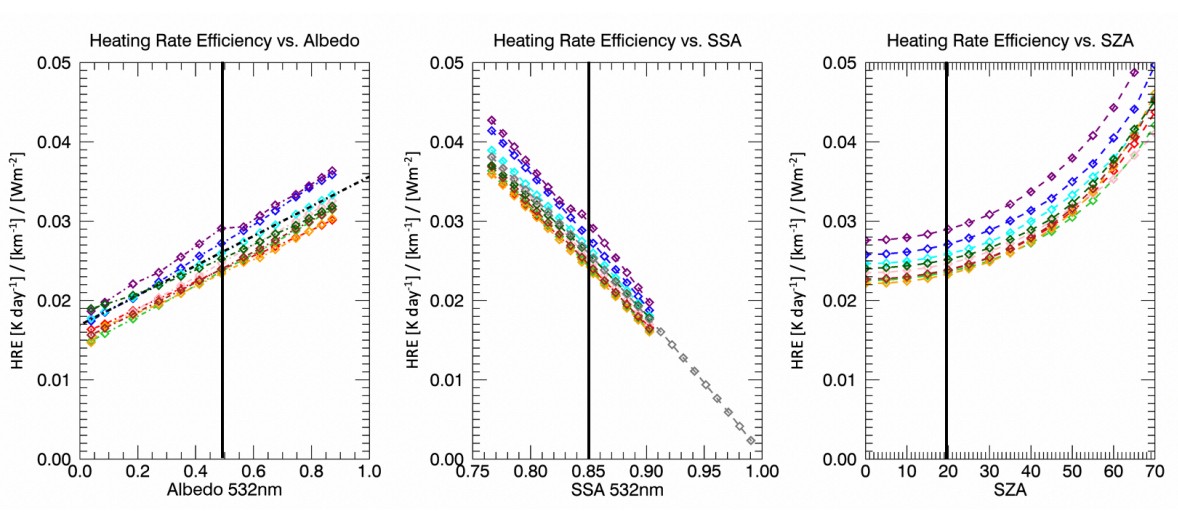


**Figure 10. The profile-median HREs for varying aerosol vertical profile s as a function of a) Albedo b) SSA and c) SZA. The individual colored sets of calculations represent the different aerosol loading and vertical distribution measured during the 9 different cases. The albedo spectra, aerosol optical properties, and SZA are consistent for each set of calculations (indicated by the black line), where the SZA for both a) and b) is set to 20 degrees, the albedo is set to 0.5 for b) and c), and the SSA is set to 0.85 for a) and c).**






| Property | Instrument(s) | Method | Reference |
|---|---|---|---|
| $SSA_\lambda$ | SSFR/4STAR | Retrieval | Cochrane et al., 2019; Cochrane et al., 2021 |
| $g_\lambda$ | SSFR/4STAR | Retrieval | Cochrane et al., 2019; Cochrane et al., 2021 |
| $AOD_\lambda$ | 4STAR | Measurement | Dunagan et al., 2013; Shinozuka et al., 2013; LeBlanc et al., 2020 |
| $Albedo_\lambda$ | SSFR | Measurement | Pilewskie et al., 2003; Schmidt and Pilewskie, 2012; Cochrane et al., 2019; Cochrane et al., 2021 |
| Aerosol extinction profile (532 nm) | HSRL | Measurement | Hair et al., 2008; Burton et al., 2018 |
| Water Vapor Profile | P-3 Hygrometer (EdgeTech Model 137 aircraft hygrometer)/4STAR | Measurement | Segal-Rosenheimer et al., 2014; Pistone et al., 2021 |
| Column Ozone | 4STAR | Measurement | Segal-Rosenheimer et al., 2014 |

**Table 1. Spiral profile and radiation wall heating rate calculation input parameters and their sources. The HSRL 532 nm extinction is used only for the radiation wall heating rates (curtains).**

| Date | SSA [501 nm] | g [501 nm] | Cloud Albedo [500 nm] | Solar Zenith Angle | AOD [500 nm] | Column water vapor [g/cm²] | Column ozone [DU] | Cloud Top Height (BOL) [km] |
|---|---|---|---|---|---|---|---|---|
| 20160831 #2 | 0.85 | 0.6 | 0.69 | 37.2 | 0.6 | 1.04 | 289.7 | 1.2 |
| 20160902 #1 | 0.81 | 0.57 | 0.6 | 28.5 | 0.42 | 1.1 | 342.3 | 1.9 |
| 20160902 #4 | 0.88 | 0.56 | 0.65 | 26.2 | 0.46 | 1.31 | 341.7 | 1.7 |





| | | | | | | | |
|---|---|---|---|---|---|---|---|
| 20160920 #1 | 0.82 | 0.56 | 0.73 | 33.8 | 0.47 | 0.87 | 410.6 | 1.0 |
| 20160920 #2 | 0.85 | 0.57 | 0.45 | 21.2 | 0.57 | 1.15 | 441.9 | 1.4 |
| 20170812 #3 | 0.84 | 0.56 | 0.57 | 46.7 | 0.32 | 1.37 | 243.8 | 1.4 |
| 20170813 #1 | 0.83 | 0.56 | 0.7 | 33.6 | 0.21 | 0.41 | 268.8 | 1.7 |
| 20170824 #1 | 0.79 | 0.56 | 0.54 | 26.4 | 0.27 | 0.77 | 326.2 | 1.1 |
| 20170830 #1 | 0.84 | 0.42 | 0.49 | 23.2 | 1.36 | 1.6 | 290.9 | 1.8 |

**Table 2 (adapted from Cochrane et al., 2020) RTM inputs for each spiral case.**

| Date | UTC | Associated Spiral (Yes/No) | SSA [532 nm] | Collocated BOL/TOL (Yes/No) | SZA range | Albedo Range [532 nm] | AOD Range [532 nm] |
|---|---|---|---|---|---|---|---|
| 20160920 | 10.9-13.5 | Yes | 0.85 | Yes | 18.5-18.9 | 0.02-0.52 | 0.09-1.0 |
| 20160924 | 10.42- 11.97 | No | 0.83 | No | 14.6-15.6 | 0.0-0.12 | 0.06-0.84 |
| 20170813 | 9.85-11.89 | Yes | 0.83 | Yes | 22.1-23.5 | 0.29-0.75 | 0.19-0.22 |
| 20170824 | 10.94-11.89 | Yes | 0.79 | Yes | 24.6-24.8 | 0.29- 0.55 | 0.05-0.42 |
| 20170826 | 12.8-14.3 | No | 0.83 | No | 37.9-40.7 | 0.07- 0.99 | 0.04-0.36 |

**Table 3. Radiation wall case information. The last 3 columns refer to the BOL leg.**

**Data Availability:**

The 2016 and 2017 ORACLES data are publicly available at https://doi.org/10.5067/Suborbital/ORACLES/P3/2016_V1 (ORACLES Science Team, 2017) and https://doi.org/10.5067/Suborbital/ORACLES/P3/2017_V1 (ORACLES Science Team, 2019).

**Author contributions**

SPC collected SSFR data, performed the bulk of the analysis, and wrote the majority of the paper with input from the other authors. KSS collected SSFR data, helped with the methodology development and data analysis, and helped with developing,



writing, and editing the paper. HC, PP, and SK helped with the data collection of SSFR. JR was one of the PIs for the ORACLES campaign and provided 4STAR data. SL was the PI of the 4STAR instrument. KP, MK, MSR, YS, and CF provided 4STAR data. RF, SB, and CH provided HSRL data. MM provided ALADIN model output. PZ was on the leadership team for

the ORACLES project and helped advise the interpretation of coincident aerosol and water vapor heating rates. All the co-authors helped in the reviewing and editing of the paper.

**Competing interests**

The authors declare that they have no conflict of interest.