# Peer review of "Biomass Burning Aerosol Heating Rates from the ORACLES 2016 and 2017 Experiments"

_Atmospheric Measurement Techniques, 2021_

## Referee Comment (RC2)

**Summary**:

This paper proposes a new approach to study heating rate profiles on the South-eastern Atlantic region, well-known for the influence of extreme biomass-burning episodes. This study is essential to understand drivers of the heating rate in this region. Even if there are some reorganisations of titles and subtitles to apply, authors well explain the method by: 1) detail the calculation of the usually analysed heating rate as well as its dependency on atmospheric absorbers, 2) analyse applications to spirals and radiation walls during the ORACLES-2016 and 2017 experiments, and 3) demonstrate the necessity of the HRE to reduce variability trained by aerosol extinction and incoming solar flux. The description of the method is precise and clear, and authors well explained the limitations of the method.

The results and analyses are interesting for aerosol impact studies in the South-eastern Atlantic region and the method may bring promising information if adapted to other area or to satellite measurements. This is why the study well fit with the journal scope and I would recommend for publication after considering following specific comments and technical corrections.

**Specific comments**:

1) The study is based on several measurement technics which allow to reduce assumptions and obtain accurate retrievals. However, combining in-situ and remote sensing measurements obtained from different flight patterns drives several issues which have to be considered to discuss the results. This is what I would expect more information and discussions on:
   - Altitudes of measurements for each case for both 4STAR and HSRL measurements. As example, does P-3 exceed 12 km height during measurements?
   - Distances and times between combined spirals and walls. This may be included on Table 3.
   - Spectral ranges and resolutions for each instrument. This may be included on Table 1.

2) The application of the method to the two experiments allows to note the stability of the HRE in the region at the biomass-burning period since the aerosol and water vapor sources are expected to be seasonal. However, as authors well explained, the method is applied at very specific atmosphere conditions and is only applicable to similar aircraft measurements conditions. In order to facilitate comparison study, I would expect more details and discussions on the parameter variations. As example:
   - Figure 6a and 6b would benefit from additional information on AOD and cloud albedo variations along the wall. Link with Figure 7c may be more discussed since it could explain the reason of aerosol heating rate variations.
   - Measured parameters on Figure 10 should be highlighted, for example with filled dots. From there, the heating rate efficiency from each spiral will be shown.
   - Figure 9 well demonstrates the stability of HRE for the two main aerosol layers between 2 and 4.5 km. More discussion on these features are expected.
   - Additional figure with the asymmetric coefficients on Figure 10 will also give relevant information of aerosol particles size impacts on HRE by considering that coarse mode particles are not well represented in the used spectral range.

**Technical corrections**:

1) Equations (2) to (5) need to be rewritten:
   - Please, detail the meaning of $(x)$.
   - In order to obtain $R_\lambda(x) = 1$ at 532 nm, equation (3) shouldn't be:
   $$AOD'_\lambda = AOD'_\lambda * \frac{AOD^{HSRL}_{532}}{AOD'^{4STAR}_{532}} \ ?$$
   - Used terms have to be consistent between equations. $AOD^{HSRL}_{532}$ in equation (5) instead of $AOD_{HSRL}$.
2) Figures 3 is discussed earlier than Figure 2 in the text. Please exchange them.
3) Figure 7 format should be consistent between panels:
   - Only mention the wall's date as title.
   - [km] on y-axis.
   - [k/day] and [Km$^{-1}$] on x-axis on Figure 7f.
4) Figure 8a values seem to correspond to cloud albedo and not SSA as Figure 8b and values on the text. Please also correct the legend.
5) Figure 8: please change y-axis with Aerosol and water vapor Heating Rate instead of on the title.
6) Please, add the coloured code used on Figure 8 and 10 on Table 2.
7) Table 2, why values are expressed at 500 nm and not at 550 or 532 nm as in the whole paper? These values have to be comparable to Table 3 values as well.
8) line 338: the sentence is not needed.

---

## Author Comment (AC1)

**Thank you for your thoughtful review of our manuscript. We appreciate your comments and questions and we have responded below (in boldface) to each item. Line numbers refer to the original manuscript.**

This is a useful analysis of the heating rates derived for biomass burning aerosols over the south-east Atlantic during the ORACLES campaign. The dependence of the heating rates on various driving factors is isolated by making use of a newly conceived parameter, the "heating rate efficiency" or HRE. This parameter removes the linear dependence on aerosol extinction coefficient and incoming solar flux in each layer (i.e. the main drivers for the heating rates) by dividing the heating due to aerosols by these factors. The remaining dependencies on underlying cloud albedo, aerosol single-scattering albedo, and path length for the direct solar beam are then readily observed.

Whilst HRE provides a useful intermediary in determining the relationship between heating rates and various parameters, I'm not convinced of its utility as a general purpose parameter (for reasons outlined in the comments below). Despite that, this is a well written and useful study that I would recommend for publication subject to fairly minor revision after addressing the following comments.

1) The method used to segregate the contribution to heating rates from different components is only really applicable for small perturbations caused by the removed component. For example, if you remove aerosol from the whole profile and then recalculate the heating rate at layers below the aerosol layer, the heating in those layers is likely to be higher than before because more direct flux would have penetrated to that depth. This would lead you to conclude the aerosol has a negative heating rate at these altitudes. These issues should be explained. A different approach would be to calculate the component's contribution to the heating rate layer-by-layer by only removing the component from a single layer each time, and therefore leaving the flux arriving at the layer principally unaltered. Was this considered?

This is a good point. We calculated the component-specific HR in two steps: 1) Total heating rate, 2) Turn off components (e.g., aerosols, water vapor, other gases) one by one, and calculate the heating rate without that component, then take the difference. This allows us to partition the HR layer by layer. Your statement is true: When turning off a component, the HR of all the *other* components located below the (now missing) layer that would normally attenuate the radiation is higher than when the component is turned on.

However, this is effect is small. The difference in downwelling irradiance at the bottom of the aerosol layer is only  $\sim 18$  W/m2 as shown in the figure below. This is less than 2% of the total irradiance at that level, and below the uncertainty in SSFR instrument (3%–5% across the spectral range) used to derive the aerosol intensive properties.

Therefore, we have chosen to keep the results as they stand but have included the following text at lines 200-201:

"While minimal for our cases, very thick absorbing aerosol layers may induce shading effects on the downwelling irradiance."

Figure 1. Downward irradiance at 532 nm for different heating rate radiative transfer calculations for case 20160920 #2. The black line shows Fdown for the full atmosphere with all components, the blue line shows Fdown for the full atmosphere without the aerosol, and the red line shows Fdown for an aerosol-only atmosphere.

2) The description of the way heating rates are segregated by absorber is a bit disjointed. The explanation of this in lines 156-166 would be better left to section 3 (along with a discussion of the issues highlighted in comment 1 above).

**We chose to present the methodology description in this way to introduce the general method of calculating heating rates (applicable to both the spiral and the wall analyses) prior to the explaining the detailed method of each type of analysis. Because of this, we would like to preserve the section layout as is. However, as noted above, we included a discussion regarding shading effects on lines 200-201.**

3) The heating rate efficiency has been defined to make use of the available observations, but these are inconsistent in wavelength: the heating rates are for the total heating over all wavelengths, the extinction is defined at 532nm and the fluxes are for the range 350 - 2100nm. The HRE parameter cannot therefore be ascribed any physical meaning and is not generally applicable, which I suspect will limit its usefulness in a wider context. You state one future use of the parameter (around line 382) would be to translate extinction profiles into heating rates. To do this you would also need not only cloud albedo, SSA and solar zenith angle but also the downward fluxes at each level between 350 - 2100nm (which is not stated). Is the intention for HRE to be used specifically with the instruments that cover the required wavelengths?

The HRE depends on the heating rate (a broadband quantity; all wavelengths integrated to obtain the broadband heating), the downward fluxes (broadband) and the extinction at 532 nm (narrowband). To translate from extinction profiles to heating rates, the first step would be to develop a full HRE parameterization, which would require the full spectra of cloud albedo, SSA, etc. However, once the parameterization has been developed than the user is only required to have these parameters at 532 nm, which are generally readily available. It may seem like a lot of required parameters, but requiring only one narrowband value is a significant improvement over requiring the full spectrum.

In addition, while you are correct that one application of the manuscript is the potential for future simplification (through the development of a full parameterization), another important application of the HRE is that it facilitates more convenient and consistent comparisons of heating rates across models, papers, study regions, etc. Currently, comparing heating rates is a major challenge due to the huge variability of contributing parameters. With the HRE, that large variability is significantly decreased.

4) The HRE parameter appears to be a fairly arbitrary intermediary parameter. It is useful to remove the linear dependence on extinction to expose the remaining dependencies. The division by downward flux, however, hides the dependence on the state of the atmosphere above as well as the solar illumination at the top of the atmosphere. Why stop there? You could have divided through by (downward + upward) flux which would have removed the dependence on the state of the underlying atmosphere (i.e. the cloud albedo), leaving the dependence on SSA and path length, etc.

In theory, this is an excellent suggestion. However, we chose not to go further in this manuscript because the relationship of HRE to the albedo is not exactly proportional. We have added text describing Figure 10 (HRE vs. Albedo) starting at line 366 to describe this more extensively:

"From the fit line, it is apparent that the HRE increases by approximately a factor of 2 across the full albedo range from 0 to 1. This can be understood intuitively, considering that the layer at an albedo of 1 is essentially illuminated "twice" – from the top and from the bottom. Of course, the heating of the layer is not exactly doubled since the illumination from the bottom (the upwelling) is less than from the top (the downwelling) due to the partial attenuation of radiation."

If we continue this work to develop a full parameterization in the future, we would need to consider the slight non-linearity of the scene albedo, in addition to other dependencies for which linearity cannot be assumed. For example, the dependence on the co-albedo (1-ssa) is only approximately linear (at ssa=1, the HRE=0 as can be seen in Figure 10b).

5) Paragraph starting at line 79 indicates HSRLs will be able to provide extinction profiles directly. Can you provide a reference / explanation for how this is done? This paragraph goes on to state that HREs could be used to translate these extinction profiles directly in to aerosol heating rates. How can this be done without knowing the downward fluxes at each level?

**We have included the following citations at line 80:**

Hair, J. W., Hostetler, C. A., Cook, A. L., Harper, D. B., Ferrare, R. A., Mack, T. L., Welch, W., Izquierdo, L. R., and Hovis, F. E.: Airborne high spectral resolution lidar for pro- filing aerosol optical properties, Appl. Optics, 47, 6734– 6752, https://doi.org/10.1364/A0.47.006734, 2008.

Hu, Y., Vaughan, M., Liu, Z., Powell, K. and Rodier, S.: Retrieving Optical Depths and Lidar Ratios for Transparent Layers Above Opaque Water Clouds From CALIPSO Lidar Measurements, IEEE Geosci. Remote Sens. Lett., 4(4), 523–526, doi:10.1109/LGRS.2007.901085, 2007. The downward flux would still need to be known at each level, but that should be a relatively straightforward radiative transfer calculation if the extinction profile is known. We admit this may not be as straightforward as we want, and that a full parameterization may need to be built without reliance on the downwelling irradiance and instead rely only on SZA, etc. Since these details are still to be explored, we didn't go further into the development of the full parameterization in this manuscript.

6) At lines 240, 292 and various other places you have used the terms 'extensive' and 'intensive'. It's not clear to me what these terms mean in this context. Could you please define your usage of these words or perhaps use an alternative description.

Extensive properties scale with the amount. Here, heating rate is an extensive property because it's proportional to extinction coefficient, which depends on number concentration of scatters and absorbers. However, if we normalize by the extinction coefficient, then HRE becomes intensive (similar to mass (extensive) vs. density(intensive.)) We have update the text starting at line 239 to the following:

"This challenge is one of the motivations for the development of the heating rate *efficiency* (Section 5) which turns the heating rate (an extensive parameter proportional to the extinction) into an intensive quantity."

Typo's and minor adjustments:

1) Abstract line 27: I think the term "curtains" needs explaining on first use.

We have updated the text on line 29 to first define a curtain as a vertical cross section.

2) Tables and figures are not referenced in order.

Thank you for pointing this out. The order of the tables and figures have been updated to be referenced in the appropriate order.

3) line 224: day^1 -> day^-1

Updated

- 4) line 337 & 338: sentence is repeated
- Deleted duplicate sentence
- 5) line 352: and -> an **Updated**

---

## Author Comment (AC3)

**Thank you for your positive review of our manuscript. We appreciate the comments and technical suggestions you provided. Below we have provided responses (in boldface) to each of the specific comments and technical corrections. Line numbers refer to the original manuscript.**

**Summary**:

This paper proposes a new approach to study heating rate profiles on the South-eastern Atlantic region, well-known for the influence of extreme biomass-burning episodes. This study is essential to understand drivers of the heating rate in this region. Even if there are some reorganisations of titles and subtitles to apply, authors well explain the method by: 1) detail the calculation of the usually analysed heating rate as well as its dependency on atmospheric absorbers, 2) analyse applications to spirals and radiation walls during the ORACLES-2016 and 2017 experiments, and 3) demonstrate the necessity of the HRE to reduce variability trained by aerosol extinction and incoming solar flux. The description of the method is precise and clear, and authors well explained the limitations of the method.

The results and analyses are interesting for aerosol impact studies in the South-eastern Atlantic region and the method may bring promising information if adapted to other area or to satellite measurements. This is why the study well fit with the journal scope and I would recommend for publication after considering following specific comments and technical corrections.

**Specific comments**:

1. The study is based on several measurement technics which allow to reduce assumptions and obtain accurate retrievals. However, combining in-situ and remote sensing measurements obtained from different flight patterns drives several issues which have to be considered to discuss the results. This is what I would expect more information and discussions on:
   - - Altitudes of measurements for each case for both 4STAR and HSRL measurements. As example, does P-3 exceed 12 km height during measurements?

   **Within the radiation walls, the 4STAR measurements are made along the BOL leg, ranging between 1.0 to 1.6 km for the cases included in our analysis. HSRL measurements are made at the TOL, which ranged between 5.1 to 7.1 km. We chose not to include this information explicitly since it was a requirement that the aerosol layer be fully captured by the HSRL measurement. For radiation walls where these criteria were not met, the case was disqualified for further analysis. The text has been updated starting at line 248 as followed to explicitly state this:**

   **"This requires a) aerosol optical properties and water vapor profile (both obtained from a spiral retrieval and held constant), b) the SSFR-measured albedo spectra, the AOD spectra and column ozone retrievals from 4STAR, all measured from the BOL leg of the radiation wall, and c) aerosol extinction profiles at 532 nm of the full aerosol layer measured by HSRL-2 from the TOL leg (Table 2). Cases for which these criteria were not met were excluded from analysis."**

o   -   Distances and times between combined spirals and walls. This may be included on Table 3.

**We have included the UTC range for the spirals in Table 2 so the times can be compared. The distance between walls and spirals is visualized in Figure 1. As noted on lines 257-258, our goal was to examine the heating rate dependence on the scene variability without knowing the albedo and AOD spectra underlying a *specific* aerosol extinction profile. We therefore assessed the dependence statistically by considering the albedo variability and the AOD.  The underlying assumption of course is that there are no changes in key parameters (aerosol intensive properties) over the timeframe**

o   -   Spectral ranges and resolutions for each instrument. This may be included on Table 1.

**The table has been updated as follows:**

| Property | Instrument(s) | Method | Reference |
|---|---|---|---|
| $SSA_\lambda$ | SSFR/4STAR | Retrieval | Cochrane et al., 2019; Cochrane et al., 2021 |
| $g_\lambda$ | SSFR/4STAR | Retrieval | Cochrane et al., 2019; Cochrane et al., 2021 |
| $AOD_\lambda$ | 4STAR (wavelength range: 350-1650 nm) | Measurement | Dunagan et al., 2013; Shinozuka et al., 2013; LeBlanc et al., 2020 |
| $Albedo_\lambda$ | SSFR (wavelength range: 350-2100 nm) | Measurement | Pilewskie et al., 2003; Schmidt and Pilewskie, 2012; Cochrane et al., 2019; Cochrane et al., 2021 |
| Aerosol extinction profile (532 nm) | HSRL (wavelengths: 355 nm, 532 nm) | Measurement | Hair et al., 2008; Burton et al., 2018 |
| Water Vapor Profile | P-3 Hygrometer (EdgeTech Model 137 aircraft hygrometer)/ 4STAR | Measurement | Segal-Rosenheimer et al., 2014; Pistone et al., 2021 |
| Column Ozone | 4STAR | Measurement | Segal-Rosenheimer et al., 2014 |

2.  The application of the method to the two experiments allows to note the stability of the HRE in the region at the biomass-burning period since the aerosol and water vapor sources are expected to be seasonal. However, as authors well explained, the method is applied at very specific atmosphere conditions and is only applicable to similar aircraft measurements

conditions. In order to facilitate comparison study, I would expect more details and discussions on the parameter variations. As example:

- o   - Figure 6a and 6b would benefit from additional information on AOD and cloud albedo variations along the wall. Link with Figure 7c may be more discussed since it could explain the reason of aerosol heating rate variations.

**Figures 6a and 6b along with their caption have been updated to include the AOD and cloud albedo information:**

[Figure]

**Figure 6. Heating rate curtains calculated using HSRL-2 measured extinction for the 20170813 radiation wall (shown here in separate plots: North (right) and South (left)). Peak heating of ~4.8 K/day occurs between 2 and 3 km. The underlying albedo, shown at 532 nm in green, is significantly higher on the left plot than the right (i.e., further south), contributing to higher aerosol heating rates. The AOD, shown at 532 nm in blue, does not vary significantly across the wall. Missing results between -7.08 and -6.51 are due to in-cloud sampling that replaced above-cloud albedo measurements and serve as the break point between North and South ends of the wall.**

- o   - Measured parameters on Figure 10 should be highlighted, for example with filled dots. From there, the heating rate efficiency from each spiral will be shown.

**While Figure 10 also uses the same color code as Figure 8, we do not want to give the impression that the HRE values are the true HRE from the spirals. Rather, Figure 10 shows a set of calculations based on the AOD and vertical distribution of the aerosol in each spiral case with constant SSA, albedo, or SZA values (depending on a, b, and c Figures). If we included filled points, they would most likely NOT align with the case-specific HRE value because when we showed the dependence of HRE on parameter (e.g., albedo), we pegged all of the other parameters (e.g., AOD, SZA) to fixed values across all cases, indicated as dashed lines in all three panels. Therefore, we do not include this label on Figure 10.**

- o - Figure 9 well demonstrates the stability of HRE for the two main aerosol layers between 2 and 4.5 km. More discussion on these features are expected.

  **Beginning at line 352, we included the following additional text in the discussion of Figure 9:**

  **"Figure 9 shows an example of the vertical profile of the aerosol heating rate, extinction, and HRE, which clearly shows that the HRE for the aerosol two sub-layers (at 2 km and 4.5 km) is rather stable, and comparable between the two layers, despite the variable extinction and heating rate profiles."**

- o - Additional figure with the asymmetric coefficients on Figure 10 will also give relevant information of aerosol particles size impacts on HRE by considering that coarse mode particles are not well represented in the used spectral range.

  **Indeed, the impact of the coarse mode on the HRE is not conveyed in these plots. However, the asymmetry parameter range was too small in our case to make meaningful statements about the size distribution. To illustrate this, we include a figure from an earlier paper (Cochrane et al., 2021) below:**

[Figure]

**Figure 7.** Critical albedo as a function of mid-visible SSA. The red dashed cross shows the case-average $\alpha_{\text{crit}}$.

**It shows that the SSA by far dominates the radiative impact of the aerosol (here visualized in terms of the critical albedo, a quantity related to the TOA radiative effect); the asymmetry parameter (labels on the individual cases) is not useful to explain any of the variability that is not explained by the SSA.**

**Technical corrections**:

1) Equations (2) to (5) need to be rewritten:

- Please, detail the meaning of $(x)$.
**x is the location, z is the height. This has been included on line 275.**

- In order to obtain $R_\lambda(x) = 1$ at 532 nm, equation (3) shouldn't be:

$$AOD'_\lambda = AOD'^{4STAR}_\lambda * \frac{AOD^{HSRL}_{532}(x)}{AOD'^{4STAR}_{532}} ?$$

**Thank you for pointing out this error. We have corrected equation (3) and re-ran the heating calculations it applies to. Figures 6 and 7 have been updated to reflect the new calculations. New Figure 6 is shown in response to comment #2. New Figure 7 shown below:**

[Figure]

- Used terms have to be consistent between equations. $AOD^{HSRL}_{532}$ instead of $AOD_{HSRL}$.
**This has been corrected.**

2) Figures 3 is discussed earlier than Figure 2 in the text. Please exchange them.
**This has been corrected.**

3) Figure 7 format should be consistent between panels:

- Only mention the wall's date as title.
- [km] on y-axis.
- [k/day] and [$Km^{-1}$] on x-axis on Figure 7f.

**Figure 7 has been updated as shown above.**

4) Figure 8a values seem to correspond to cloud albedo and not SSA as Figure 8b and values on the text. Please also correct the legend.

**To convey as much information as possible, Figure 8a points are intentionally labeled by their corresponding 550 nm albedo values while Figure 8b points are labeled by the SSA (described in the figure caption.) The legend indicates that the dashed line is an additional calculation performed using the mean SSA from all cases. The text has been corrected at line 328.**

5) Figure 8: please change y-axis with Aerosol and water vapor Heating Rate instead of on the title.

**Figure 8 has been updated as follows:**

[Figure]

**Figure 8. a) Aerosol heating rate as a function of AOD at 550 nm. Column-averaged values from each spiral case are shown as colored points labeled by their 550 nm albedo value. The black dashed line indicates RTM calculations using mean SSA (0.83, 550 nm) and albedo (0.6, 550 nm) from all cases, and a range of AOD spectra (ranging from 0 to 1.4 at 550 nm). b) Water vapor heating rate as a function of the water vapor path. The grey dashed line is a simple linear fit to highlight the dependence. Cases are labeled by the 550 nm SSA value. The color-coding in both a) and b) is denoted by the legend on a).**

6) Please, add the coloured code used on Figure 8 and 10 on Table 2.

**The color code has been added to Figure 8 as shown above. While Figure 10 also uses the same color code, we do not want to give the impression that the HRE values are the true HRE from the spirals. Rather, Figure 10 shows a set of calculations based on the AOD and vertical distribution of the aerosol in each spiral case with constant SSA, albedo, or SZA values (depending on a, b, and c Figures). If we included filled points, they would most likely NOT align with the case-specific HRE value because when we showed the dependence of HRE on parameter (e.g., albedo), we pegged all of the other parameters (e.g., AOD, SZA) to fixed values across all cases, indicated as dashed lines in all three panels. Therefore, we do not include this label on Figure 10.**

7)  Table 2, why values are expressed at 500 nm and not at 550 or 532 nm as in the whole paper? These values have to be comparable to Table 3 values as well.

**Table 2 was adapted from the Cochrane et al., 2021 paper in which the optical properties were retrieved for each of the spiral cases. We have updated the table to report the 532 nm values.**

8)  line 338: the sentence is not needed.

**We would prefer to keep this sentence as it highlights the importance of water vapor heating in conjunction with aerosol heating.**

---

## Author Response (AR2)

Thank you, Jim, for your review and conditional acceptance of the manuscript. Below we list the additional edits requested by Reviewer #3.

**Technical point 1:**

In response to the comments of reviewer #3 and the editor regarding our initial statement on line 83-84: "The HSRL-derived extinction profiles could be directly translated into aerosol heating rates for regions where HRE is available, bypassing, and thus directly constraining, radiative transfer calculations.", we updated the text to "It is possible that the HSRL-derived extinction profiles could be directly translated into aerosol heating rates for regions where HRE and downwelling irradiance are available." where we also added the caveat that the reviewer and editor both point out. We acknowledge that this may limit the practical use of HRE. However, in a previous response, we showed that the sensitivity to changes in the downwelling irradiance is small. In addition, the real power of the HRE is in cross-comparing HRE across regions and data sets more efficiently.

Similarly, we included additional text at lines 392-393 to reflect the concern: "This small variability shows HRE could be used to translate extinction profiles in the region directly into aerosol heating rates if mid-visible cloud albedo and SSA are also known. In other words, the variability in extensive parameters (e.g., extinction) is higher than intensive parameters (e.g., SSA, g) and therefore, regionally and seasonally defined HRE are useful. If available for a specific region, the HRE concept would allow a direct translation from mid-visible extinction to heating rate, provided that the downward irradiances are available either through observations or radiative transfer calculations. Of course, if SSA varies appreciably within the layer, that dependence may have to be made explicit. Alternatively, if in the future the absorption coefficient were available at sufficient accuracy in addition to the extinction coefficient, the HRE could be redefined to normalize by the absorption coefficient, thereby accounting for the SSA vertical dependence."

**Technical Point 2:**

To address the concern regarding the discussion of the heating rate calculation technique limitations, we included the following text at lines 200-201 per the reviewer's suggestion: "This technique is appropriate where the removed component introduces a small perturbation to the downward flux, such as the cases presented here. However, very thick absorbing aerosol layers may induce shading effects on the downwelling flux, leading to a low bias in the calculated heating rates towards the bottom of the aerosol layer. This effect is minimal for our cases, but a modified technique should be considered for optically thick aerosol layers."